# Healthy Lifestyle Related to Executive Functions in Chilean University Students: A Pilot Study

**DOI:** 10.3390/healthcare12101022

**Published:** 2024-05-15

**Authors:** Felipe Caamaño-Navarrete, Carlos Arriagada-Hernández, Gerardo Fuentes-Vilugrón, Lorena Jara-Tomckowiack, Alvaro Levin-Catrilao, Pablo del Val Martín, Flavio Muñoz-Troncoso, Pedro Delgado-Floody

**Affiliations:** 1Physical Education Career, Faculty of Education, Universidad Autónoma de Chile, Temuco 4780000, Chile; felipe.caamano@uautonoma.cl (F.C.-N.); carlos.arriagada@uautonoma.cl (C.A.-H.); gerardo.fuentes@uautonoma.cl (G.F.-V.); 2Collaborative Research Group for School Development (GICDE), Temuco 4780000, Chile; lorenajarat@gmail.com; 3Faculty of Education, Universidad Católica de Temuco, Temuco 4780000, Chile; 4Doctoral Programme in Physical Activity Sciences, Faculty of Education Sciences, Universidad Católica del Maule, Talca 3460000, Chile; alvaro.levin@uautonoma.cl; 5Chilean Observatory of Physical Education and School Sport, Faculty of Education and Social Sciences, Universidad Andres Bello, Las Condes, Santiago 7550000, Chile; pablo.delval@unab.cl; 6Faculty of Social Sciences and Arts, Universidad Mayor, Temuco 4780000, Chile; flaviomunoz@gmail.com; 7Department of Psychology and Anthropology, Faculty of Education and Psychology, Universidad de Extremadura, 06071 Badajoz, Spain; 8Department of Physical Education, Sport and Recreation, Universidad de La Frontera, Temuco 4811230, Chile

**Keywords:** executive function, lifestyle, physical activity, screen time, students

## Abstract

Background: A negative lifestyle is reported to be related to cognitive problems. However, there is little information about this in relation to university students. The objective of the present study was to investigate the association between executive functions (EFs) and lifestyle parameters (i.e., physical activity (PA), sleep duration, screen time (ST), and food habits) among Chilean university students. Methods: This cross-sectional study included a total of 150 university students (94 females and 56 males, aged 21.28 ± 3.15 and 22.18 ± 2.90 years, respectively). Cognitive outcomes were measured using the CogniFit assessment battery. Lifestyle was measured through validated questionnaires. Results: Across the total sample, attention exhibited a positive association with PA h/week (β: 24.34 95% CI: 12.46 to 36.22, *p* = 0.001). Additionally, coordination was positively associated with PA h/week (β: 15.06 95% CI: 0.62 to 29.50, *p* < 0.041). PA h/week was positively linked with reasoning (β: 20.34 95% CI: 4.52 to 36.17, *p* = 0.012) and perception (β: 13.81 95% CI: 4.14 to 23.49, *p* = 0.005). Moreover, PA h/week was significantly linked to memory (β: 23.01 95% CI: 7.62 to 38.40, *p* = 0.004). In terms of the EFs, PA h/week showed a positive association with cognitive flexibility (β: 45.60 95% CI: 23.22 to 67.69, *p* = 0.001). Conclusions: In conclusion, lifestyle (PA h/week) was positively associated with EFs. Therefore, an increase in PA levels among these students should be a target for community- and university-based interventions in order to promote cognitive development such as attention, coordination, reasoning, perception, memory, and cognitive flexibility.

## 1. Introduction

Executive functions (EFs) refer to a set of higher-order cognitive abilities that enable the assessment and achievement of a goal [1]. In addition, EFs are fundamental for self-regulation, problem solving and decision making [2]. Furthermore, it has been suggested that EFs are essential for developing adaptive behaviours that involve diverse ways of processing information, different sensory modalities, and executing responses, including aspects related to memory and emotional regulation [3]. Moreover, higher EFs among university students have been associated with various benefits [2,4,5]. Complementary to the above, EFs exhibit varying profiles depending on the developmental stage of humans. These functions begin to emerge in infancy with basic skills (up to the age of 3), and more specific skills develop during early childhood. They continue to develop at their own pace in adolescence [6], and cognitive performance peaks during young adulthood [6,7]. However, they decline in old age, mainly due to structural and functional changes in the prefrontal cortex [8]. 

Existing evidence suggests that university is a crucial stage for the development of EFs [9]. In this sense, in a university context, EFs have a significant impact on student success and achievement [10]. In this sense, previous evidence has shown that EFs are linked to academic success, indicating the importance of promoting learning and achievement [5,11,12]. However, EFs not only contribute to the academic performance of students but are also related to mental health, physical health, health-related quality of life (HRQoL), and job success [2]. Complementary to the above, recent studies have shown that EFs could be related to health levels, lifestyle parameters (i.e., physical activity [PA] levels, diet quality, and sleep quality), and emotional regulation [5,13,14,15,16]. Complementary to the above, EFs may play a potential role in social functioning in university students [17]. On the other hand, university students who show poorer levels of EFs can be expected to have problems in the study process [18]. Therefore, addressing poor EFs as a variable for measurement among university students could be considered a priority. 

Unhealthy lifestyles have become a public health concern [19] and are associated with increased cardiovascular morbidity and mortality [20]. In addition, university students form part of the population most at risk of developing unhealthy lifestyle behaviours [21]. Moreover, it has been indicated that the university context is a critical stage during which students begin making their own decisions [22]. When individuals enter university, they experience changes associated with increased autonomy and exposure to a new environment. This can lead to higher levels of stress, which may result in an increase in unhealthy patterns [19,23,24,25]. In this context, it has been demonstrated that university students often have poor health habits [26]. Evidence has also shown that university students are more prone to unhealthy lifestyle choices (i.e., physical inactivity, sedentary behaviour, unhealthy eating habits, smoking, and alcohol consumption) [27,28]. Likewise, healthy lifestyle patterns can improve overall health, as well as preventing disease [29]. In line with the above, university students may be more prone to weight gain due to extended periods of screen time (ST) on devices such as mobile phones and computers. This can make it challenging for them to find the motivation to engage in activities that promote healthy living [23]. 

People’s lifestyles have commonly been investigated in the context of health; however, it has been shown that lifestyle factors are intertwined with EFs in university students [30]. For instance, research has shown that engaging in physical activities, such as sports, has a positive impact on brain health, improves cognitive functions, and reduces the risk of dementia in old age [9]. Complementarily, healthy lifestyle habits such as healthy eating habits and practising PA could offer protection against cognitive decline [31]. Similarly, ST activities may be negatively linked with EFs [32]. This statement demonstrates a clear connection between leading a healthy lifestyle and the cognitive abilities of humans. Likewise, previous data regarding young people have indicated that unhealthy lifestyles (i.e., PA levels and poor sleep quality) were linked with poorer EFs [33]. Moreover, a previous study conducted with Mexican university students indicated that EFs were positively related with healthy lifestyle factors such as eating habits [34]. Furthermore, it has been shown that better PA levels are linked with higher executive inhibitory control especially in female university students [9]. Complementary to the above, it has been reported that more frequent moderate-to-vigorous or light PA was related to better EFs in young adults [35]. Indeed, existing evidence shows a general consensus that developing healthy lifestyle habits such as PA is helpful for cognitive functioning [36]. In a complementary way, a recent study conducted among college students reported that increased daily participation in PA could be beneficial for their EFs and, in addition, PA was negatively associated with negative emotion [37]. Therefore, the evidence contributes to consolidating the positive association between PA and brain function [38]. In this context, another investigation showed that PA and exercise may have positive effects on cognitive processes [39]. On the other hand, an unhealthy lifestyle may negatively impact EFs. In this context, it has been indicated that poor and unhealthy habits (i.e., skipping breakfast) could impact negatively on cognitive functions [40]. Intriguingly, a pilot study conducted in university students reported that sedentary behavior negatively predicted cognitive inhibition [41], and it has been shown that better EFs are linked with less sedentary behavior in university students [42]. The above is important to consider since it has been reported that university promotes behaviors such as sitting [43]. 

Against this background and to the best of our knowledge, no other study has explored the association between EFs and lifestyle in Chilean university students. The objective of this study was, therefore, to investigate the association between EFs and lifestyle parameters (i.e., PA, sleep duration, ST, and food habits) among Chilean university students and to determine the differences in EFs and lifestyle parameters according to sex. 

## 2. Materials and Methods

A descriptive study with a cross-sectional design was developed. A total of 150 university students (94 females and 56 males, aged 21.28 ± 3.15 and 22.18 ± 2.90 years, respectively) from the Universidad Autónoma de Chile in Temuco, Chile participated in the study. A total of 33 subjects were excluded (women not meeting the inclusion criteria or for other reasons (*n* = 20); men not meeting the inclusion criteria or for other reasons (*n* = 13). The sample was intentional and non-probabilistic by convenience. 

The inclusion criteria encompassed the following conditions: (i) obtaining informed consent from the participants and (ii) being university students. The exclusion criteria were as follows: (i) any musculoskeletal injuries or medical contraindications that would prevent subjects from performing averagely in the assessments and (ii) not being present at the time of the evaluations or failing to provide informed consent. The investigation complied with the Declaration of Helsinki (2013) and was approved by the Ethics Committee of the Universidad Autónoma de Chile, Chile (N° CEC 18-23 Act). All students gave their written consent on the day of the assessment.

### 2.1. Measures 

#### 2.1.1. Cognitive Battery

To determine the cognitive domains and EFs, the 40 min CogniFit (San Francisco, CA, USA) neurocognitive assessment battery was used [31,32]. In this study, apart from the cognitive dimension score (i.e., attention, perception, reasoning, coordination, and memory), EFs (i.e., inhibition, working memory, and cognitive flexibility) were examined. Correspondingly, this battery has been reported to have good reliability [32]. In addition, this cognitive battery was previously performed by adult subjects [33].

The application of this evaluation is simple and intuitive, so that any professional can apply it without difficulty. In addition, it has been designed so that it can be used either face to face in a consultation or laboratory or remotely from the homes of patients or participants. This neuropsychological test was administered online, with an approximate duration of 30–40 min. At the end of the evaluation, a complete results report was automatically obtained with the user’s neurocognitive profile. In addition, this testing method provides valuable information that, as professionals, can help us detect whether there is a risk of any disorder or problem, recognize its severity, and identify the most appropriate support strategies for each case. It is recommended that this neuropsychological assessment is performed when the researcher wants to know the brain function or cognitive, physical, psychological, or social well-being of the patient or participant. This evaluation battery should be used complementary to the professional diagnosis and not as a substitute for a clinical interview. Each of the neuropsychological tasks contained in the cognitive assessment battery (CAB) for professionals has been validated following the scientific method. This ensures appropriate psychometric characteristics for an effective evaluation of the patient’s or participant’s cognitive status. The cognitive profile of the neuropsychological report has high reliability, consistency, and stability. Through cross-sectional research designs, psychometric statistics have been obtained with values close to 0.9, such as Cronbach’s alpha coefficient. The test–retest tests have obtained values close to 1, which demonstrates high reliability and precision [32,33].

#### 2.1.2. Lifestyle

To evaluate the quality of the university students’ diet, students completed a diet questionnaire that had previously been used with Chilean university students [44]. The instrument consisted of 15 dichotomous-response questions (yes–no) about eating habits. The scores were categorized as follows:

≥13 points = healthy eating; between 10 and 12 points = you are on the right track, but you should improve; between 7 and 9 points = unhealthy eating; and ≤6 points = very unhealthy eating [45].

The original survey classified respondents into “healthy eating” (≥13 points), “you are on the right track, but you need to improve” (between 10 and 12 points), “unhealthy eating” (between 7 and 9 points), and “unhealthy diet” (≤6 points); to facilitate data analysis, the first two categories were merged. The final instrument was subjected to validation through expert judgment, carried out by two nutritionists. In addition to the survey, the place of residence and daily mealtimes on campus were considered.

PA was determined by using a short version of the International Physical Activity Questionnaire [46]. This instrument has been used and validated in Chilean adults [47]. The questions ask about the time spent being physically active in the past 7 days. It requests the following from the participants: Please answer each question even if you do not consider yourself an active person. Please think about the activities you do in study time or work, as part of your home or garden tasks, moving from one place to another, or in your free time for recreation, exercise, or sport. Think about all the intense activities you did in the past 7 days. Intense physical activities refer to those that involve intense physical effort and make you breathe much harder than normal. Think only about those physical activities that you did for at least 10 min straight.

Screen time and sleep duration were determined through the following questions that had been previously used in different studies [48,49]: “How many hours a week do you watch videos?”, “How many hours a week do you play video or computer games?”, and “How many hours of sleep do you usually get per day and/or night?”. The ST was computed from the sum of the two questions and analyzed in hours/day. Complementary to the above, a study conducted with a nationwide sample of youth indicated that the questions on ST and sleep are applicable to a young population [50].

The questionnaires and the CogniFit neurocognitive assessment battery were completed individually in the presence of assistant researchers (who respected data confidentiality and clarified any potential doubts or questions). All the evaluations took place in a computer lab during the morning.

### 2.2. Statistical Analysis

Statistical analyses were performed using SPSS version 21.0 (SPSS Inc., Chicago, IL, USA). The Kolmogorov–Smirnov test and Levene’s test were used to assess the normal distribution of data and homogeneity of variances. Continuous variables were expressed as means and confidence intervals. Differences in the comparison between the sexes were established using an analysis of variance Student’s *t*-test. To determine the association between EFs and lifestyle parameters, a simple linear regression was used. The significance level was set at *p* < 0.05.

## 3. Results

Table 1 displays a comparison of the study variables according to sex. Significant differences in attention (*p* = 0.011), working memory (*p* = 0.037), diet quality score (*p* = 0.037), PA h/week (*p* = 0.001), ST h/day (*p* = 0.012), and sleep duration h/day (*p* = 0.036) were observed.

In the total sample, attention exhibited a positive association with PA h/week (β: 24.34 95% CI: 12.46 to 36.22, *p* < 0.001). Additionally, coordination was positively associated with PA h/week (β: 15.06 95% CI: 0.62 to 29.50, *p* = 0.041) (Table 2).

Conversely, PA h/week was positively linked with reasoning (β: 20.34 95% CI: 4.52 to 36.17, *p* = 0.012) and perception (β: 13.81 95% CI: 4.14 to 23.49, *p* = 0.005) (Table 3).

Moreover, PA h/week was significantly linked to memory (β: 23.01 95% CI: 7.62 to 38.40, *p* = 0.004) (Table 4).

In terms of executive functions, PA h/week showed a positive association with cognitive flexibility (β: 45.60 95% CI: 23.22 to 67.69, *p* = 0.001) (Table 5).

## 4. Discussion

The objective of this study was to investigate the association between EFs and lifestyle parameters (i.e., PA, sleep duration, ST, and food habits) among Chilean university students and to determine the differences in EFs and lifestyle parameters according to sex. The main findings of this study are as follows: (i) men had better results in attention, memory, working memory, diet, and PA h/week. Women had less ST h/day and more hours of sleep; (ii) PA h/week was positively related to attention, coordination, reasoning, perception, memory, and cognitive flexibility; and (iii) ST, food habits, and sleep were not associated with EFs. 

These findings highlight the importance of considering gender differences when analyzing EFs and lifestyle habits among university students. In this regard, another study among university students showed that there were differences in self-regulatory EFs according to sex [51]. Moreover, a cross-sectional study reported that EFs were linked with the orbitomedial cortex and moderated by sex in university students [52]. A recent study indicated that there were no differences in EFs between men and women [53]. Furthermore, a previous study among university students showed that men had higher PA scores than women [54]. In addition, it has been reported that female students spend more time studying [55]. In line with the above, previous evidence showed that women presented a higher proportion of physical inactivity than men among university students [56]. Other evidence has suggested that males had betters results on working memory tasks than females, while females had better results in reading comprehension than their counterparts [57]. Likewise, a previous investigation indicated that the executive functioning was related to individual differences such as sex [58]. Moreover, a systematic review with meta-analysis showed sex differences in verbal working memory [59]. Complementary to the above, specific gender differences in cognitive tasks have been reported [60]. Likewise, another study found that there were sex differences in selective attention [61]. In addition, it was found that there were sex differences in inhibitory control among university students [62]. Complementary to the above, another study reported sex differences in EFs in university students [63]. 

In this study, we found that PA h/week was positively related to cognitive flexibility. This evidence contributes to consolidating the positive association between PA and EFs in university students. In this sense, the positive links between PA and EFs observed in this study are consistent with extensive sections of the literature [64,65,66,67,68]. In this context, a systematic review reported that PA can be a way to improve cognitive outcomes and language skills in adolescents and young adults [69]. Another study indicated that EFs and PA could influence academic performance [70]. Similarly, a cross-sectional study among university students reported that PA was positively related to EFs [9]. In agreement with the above, a study indicated that practicing regular PA could have a beneficial and multifaceted impact on executive functioning, which encompasses various cognitive areas that are crucial for academic performance and daily functioning [71]. Likewise, PA has been shown to improve general cognitive functioning, including attention, memory, cognitive flexibility, and problem-solving skills [72]. In this regard, a study conducted with university students found that vigorous PA was linked with inhibition and working memory [73]. Another study conducted in adult subjects showed a positive link between objectively measured PA (i.e., moderate-to-vigorous and light physical activity) and EFs [35]. In addition, it has been shown that regular PA has a positive selective impact on EFs in adult subjects [74]. Moreover, data from university students showed that PA was associated with subjective EFs [75]. Therefore, increasing university students’ daily PA could positively influence EFs. 

In this context, another study among university students found that healthy lifestyles (including PA) were positive for inhibitory control performance [76]. In addition, it has been reported that acute aerobic PA positively impacts inhibitory control [77]. Data from female college students showed that PA was positively related to working memory [78]. This previous evidence contributes to solidifying the positive association between PA and EFs in university students. Complementary to the above, a recent study concluded that increasing PA could improve working memory performance in college students [79]. Building on previously reported findings, it is suggested that during university, exercise may be linked to better cognitive flexibility [80]. Similarly, another study indicated that PA was positively associated with EFs, specifically in relation to executive inhibitory control [9]. Likewise, there is strong evidence about the positive relation between being active and EFs in university students [35]. Consistent with the above, another study has shown that PA is linked to better EFs and increased EFs may improve academic performance in university students [81]. 

In this study, we found that PA h/week was positively related to attention, coordination, reasoning, perception, and memory. In this line, it has been shown that practice PA and exercise may have positive effects on cognitive processes [39]. For example, a study reported a positive association between healthy lifestyle that included PA and cognitive function such as selective attention and concentration [82]. In this context, previous evidence indicated that PA is linked with better attention and memory [83]. Likewise, it has been indicated that PA promotes the release of growth factors and reduces brain inflammation; therefore, PA could prevent cognitive decline [68]. In addition, another study reported that healthy lifestyle habits were related to better cognitive function [84]. In addition, previous evidence has shown that PA is an effective method to stimulate brain plasticity [85]. In this context, a study among university students reported that PA was positively related to cognitive functions such as creativity [86]. Moreover, it has been reported that PA may improve attention and psychomotor vigilance in undergraduate students [87]. For example, a recent study reported that a single session of aerobic exercise increased BDNF serum levels [88]. Complementary to the above, PA interventions have been shown to improve EFs in university students [89]. However, that study did not find associations between ST, eating habits, sleep, and EFs. In this context, data from three empirical studies reported no relation between sleep quality and EFs in young adults [90]. Contrary to our results, another study reported that better EFs were related to less follow-up sedentary behavior [42]. Additionally, it has been reported that healthy eating habits are related to EFs among university students [30]. Previous evidence has shown that a healthy lifestyle is beneficial for EFs [91]. In addition, another study among university students reported that healthy eating habits were positively associated with academic performance [92]. Moreover, it has been indicated that a healthy lifestyle may decrease the risk of cognitive decline [93]. 

In the present study, the main limitation is the cross-sectional design. In addition, we used a convenience sample. Among the study’s strengths, we can highlight (i) the simplicity of the assessments (which would allow their use and application in healthy lifestyle interventions focused on university students) and (ii) the fact that cognitive measures were obtained using a computer neurocognitive assessment battery. 

## 5. Conclusions

In conclusion, firstly, attention exhibited a positive association with PA h/week. This suggests that engaging in regular PA is beneficial for maintaining attentional processes among university students, potentially enhancing their ability to focus and concentrate during academic tasks and activities. Secondly, coordination was also positively associated with PA h/week. This finding implies that physical activities improve coordination skills. Interestingly, this study also found that PA h/week was positively linked with reasoning and perception. This suggests that engaging in PA may benefit not only physical health but also cognitive processes related to logical thinking, problem-solving, and sensory perception. This is particularly important in a university setting where students are often required to engage in complex cognitive tasks and academic challenges. Moreover, the association between PA h/week and memory highlights the potential cognitive benefits of PA. Regular engagement in physical activities may support memory functions, including encoding storage and retrieval of information. This has implications for academic performance, as effective memory abilities are crucial for learning and retaining new information. In terms of EFs, this study found a positive association between PA h/week and cognitive flexibility. EFs play a critical role in higher-order cognitive processes such as planning, decision making, and adaptability to changing situations. The positive association with cognitive flexibility suggests that PA may enhance students’ ability to switch between different tasks, strategies, or mental sets, which are essential skills for academic and real-life success. Likewise, this study underscores the importance of lifestyle factors such as PA in promoting cognitive performance, specifically EFs, among university students. These findings highlight the multidimensional benefits of PA beyond physical health, extending to cognitive functions that are crucial for academic achievement and overall well-being. Therefore, promoting and encouraging regular PA among university students can serve as a valuable strategy to enhance cognitive performance and support academic success. Future research and interventions may further explore the mechanisms underlying the relationship between physical activity and cognitive functions to develop targeted strategies for optimizing cognitive outcomes in educational settings. This conclusion is drawn from the positive associations observed between PA h/week and attention, coordination, reasoning, perception, memory, and cognitive flexibility. Therefore, an increase in PA levels among these students should be a target for community- and university-based interventions in order to promote cognitive development such as attention, coordination, reasoning, perception, memory, and cognitive flexibility. 

## Figures and Tables

**Table 1 healthcare-12-01022-t001:** Comparison of study variables according to sex.

	Total (*n* = 150)	Female (*n* = 94)	Male (*n* = 56)	*p*-Value	F-Value
Age (y)	21.61 ± 3.08	21.28 ± 3.15	22.18 ± 2.90	0.083	3.05
Cognitive outcomes					
Attention (score)	471.1 ± 124.40	450.45 ± 123.84	506.62 ± 118.31	0.011	6.72
Coordination (score)	362.61 ± 145.54	346.64 ± 153.04	390.08 ± 128.49	0.093	2.86
Reasoning (score)	397.13 ± 162.12	384.91 ± 169.76	418.16 ± 147.34	0.250	1.33
Memory (score)	294.19 ± 158.79	270.34 ± 146.20	335.22 ± 172.26	0.021	5.45
Perception (score)	343.91 ± 98.28	335.94 ± 97.26	357.62 ± 99.49	0.216	1.54
Inhibition (score)	406.90 ± 233.12	410.92 ± 241.62	400.00 ± 219.91	0.793	0.07
Working memory (score)	239.68 ± 191.84	213.56 ± 171.72	284.62 ± 216.75	0.037	4.45
Cognitive flexibility (score)	426.13 ± 235.00	410.01 ± 241.97	453.86 ± 222.14	0.296	1.10
Lifestyle parameters					
Diet quality (score)	7.37 ± 3.17	6.96 ± 3.00	8.07 ± 3.36	0.037	4.41
Physical activity (h/week)	2.01 ± 1.76	1.66 ± 1.38	2.61 ± 2.14	0.001	10.83
Screen time (h/day)	2.63 ± 0.78	2.51 ± 0.71	2.84 ± 0.84	0.012	6.44
Sleep (h/day)	6.89 ± 1.52	6.93 ± 1.61	6.81 ± 1.36	0.036	0.23

The data are presented as means and standard deviations, with statistical significance set at *p* < 0.05.

**Table 2 healthcare-12-01022-t002:** Association of cognitive dimension scores with lifestyle variables in university students.

Outcomes	Attention			Coordination		
	Beta (95% CI)	*p*-Value	Standardized Beta (SE)	Beta (95% CI)	*p*-Value	Standardized Beta (SE)
Lifestyle parameters						
Diet score	−5.74 (−12.35; 0.86)	*p* = 0.088	−0.14 (3.34)	−6.40 (−14.44; 1.62)	*p* = 0.117	−0.13 (4.06)
Physical activity (h/day)	24.34 (12.46; 36.22)	*p* = 0.001	0.35 (6.00)	15.06 (0.62; 29.50)	*p* = 0.041	0.18 (7.29)
Screen time (h/day)	0.48 (−25.34; 26.31)	*p* = 0.971	−0.03 (13.23)	12.53 (−18.85; 43.93)	*p* = 0.431	0.06 (15.86)
Sleep duration (h/day)	−3.41 (−16.67; 9.83)	*p* = 0.611	−0.04 (6.70)	−3.21 (−19.32; 12.89)	*p* = 0.694	−0.03 (8.14)

The data shown represent beta (95% CI), standardized beta, and standard error (SE). Values of *p* < 0.05 were considered statistically significant. Model was adjusted for age.

**Table 3 healthcare-12-01022-t003:** Association of cognitive dimensions score with lifestyle variables in university students.

Outcomes	Reasoning			Perception		
	Beta (95% CI)	*p*-Value	Standardized Beta (SE)	Beta (95% CI)	*p*-Value	Standardized Beta (SE)
Lifestyle parameters						
Diet score	−8.91 (−17.72; −0.11)	*p* = 0.047	−0.17 (4.45)	1.32 (−4.06; 6.70)	*p* = 0.628	0.04 (2.72)
Physical activity (h/day)	20.34 (4.52; 36.17)	*p* = 0.012	0.22 (8.00)	13.81 (4.14; 23.49)	*p* = 0.005	0.25 (4.89)
Screen time (h/day)	12.86 (−21.54; 47.27)	*p* = 0.461	0.06 (17.39)	1.87 (−19.15; 22.91)	*p* = 0.860	0.01 (10.63)
Sleep duration (h/day)	−10.12 (−27.78; 7.53)	*p* = 0.259	−0.09 (8.92)	−5.80 (−16.60; 4.98)	*p* = 0.289	−0.09 (5.45)

The data shown represent beta (95% CI), standardized beta, and standard error (SE). Values of *p* < 0.05 were considered statistically significant. Model adjusted for age.

**Table 4 healthcare-12-01022-t004:** Association of cognitive dimensions score with lifestyle variables in university students.

Outcomes	Memory			Inhibition		
	Beta (95% CI)	*p*-Value	Standardized Beta (SE)	Beta (95% CI)	*p*-Value	Standardized Beta (SE)
Lifestyle parameters						
Diet score	−2.24 (−10.80; 6.32)	*p* = 0.605	−0.04 (4.33)	−8.57 (−21.45; 4.30)	*p* = 0.190	−0.11 (6.50)
Physical activity (h/day)	23.01 (7.62; 38.40)	*p* = 0.004	0.26 (7.78)	21.54 (−1.60; 44.68)	*p* = 0.068	0.16 (11.69)
Screen time (h/day)	20.55 (−12.91; 54.02)	*p* = 0.226	0.10 (16.91)	16.71 (−33.60; 67.02)	*p* = 0.512	0.05 (25.43)
Sleep duration (h/day)	−13.80 (−30.98; 3.36)	*p* = 0.114	−0.13 (8.68)	−15.98 (−41.80; 9.83)	*p* = 0.223	−0.10 (13.04)

The data shown represent beta (95% CI), standardized beta, and standard error (SE). Values of *p* < 0.05 were considered statistically significant. Model adjusted for age.

**Table 5 healthcare-12-01022-t005:** Association of cognitive dimension scores with lifestyle variables in university students.

Outcomes	Working Memory			Cognitive Flexibility		
	Beta (95% CI)	*p*-Value	Standardized Beta (SE)	Beta (95% CI)	*p*-Value	Standardized Beta (SE)
Lifestyle parameters						
Diet score	3.61 (−7.15; 14.37)	*p* = 0.508	0.05 (5.44)	−4.14 (−16.59; 8.31)	*p* = 0.512	−0.05 (6.29)
Physical activity (h/day)	13.56 (−5.78; 32.90)	*p* = 0.168	0.12 (9.77)	45.60 (23.22; 67.99)	*p* = 0.001	0.34 (11.31)
Screen time (h/day)	5.78 (36.27; 47.84)	*p* = 0.786	0.02 (21.25)	1.13 (47.54; 49.80)	*p* = 0.963	0.00 (24.60)
Sleep duration (h/day)	−8.71 (−30.29; 12.87)	*p* = 0.426	−0.07 (10.90)	−17.27 (−42.24; 7.70)	*p* = 0.174	−0.11 (12.62)

The data shown represent beta (95% CI), standardized beta, and standard error (SE). Values of *p* < 0.05 were considered statistically significant. Model adjusted for age.

## Data Availability

Data are contained within the article.

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
