# Peer review of "Healthy Lifestyle Related to Executive Functions in Chilean University Students: A Pilot Study"

_healthcare, 2024, doi:10.3390/healthcare12101022_

Round 1

Reviewer 1 Report

Comments and Suggestions for Authors

This article addresses a current and relevant topic by examining the relationship between lifestyle factors, physical activity, and cognitive performance among Chilean students. This topic is of significant interest to the medical and educational communities due to the growing understanding of the impact of lifestyle on mental and cognitive health, given that students are often exposed to stress. The article is quite well structured and has all the necessary sections. The presentation is coherent and simple, making it easy to understand the aims, results and implications of the study. However, the discussion could have been more interesting from a more extensive critical analysis. The study is supported by a good factual base, using proven questionnaires and cognitive tests. At the same time, however, the sample size is relatively small. Statistical methods are applied appropriately, with regression coefficients, confidence intervals and significance levels clearly stated. The analysis takes into account multiple variables to provide a comprehensive view of the data. The discussion integrates the findings with existing contemporary literature, highlighting the study's contribution to understanding the effects of physical activity on cognitive function. The conclusions are drawn taking into account some of the limitations of this study. Thus, the study presented in the article is well-conducted and makes a significant contribution to understanding the relationship between lifestyle factors and cognitive function among university students. In the future, it would be appropriate to apply methodological approaches that will enhance the reliability and practical applicability of the results.

Author Response

Reviewer 1

This article addresses a current and relevant topic by examining the relationship between lifestyle factors, physical activity, and cognitive performance among Chilean students. This topic is of significant interest to the medical and educational communities due to the growing understanding of the impact of lifestyle on mental and cognitive health, given that students are often exposed to stress. The article is quite well structured and has all the necessary sections. The presentation is coherent and simple, making it easy to understand the aims, results and implications of the study.

Response: Dear reviewer, thank you very much for your comment.

 However, the discussion could have been more interesting from a more extensive critical analysis. The study is supported by a good factual base, using proven questionnaires and cognitive tests.

Response: thank you for your comment, we have added information in all introduction.

 In additions, it has been found that there were sex differences in inhibitory control among university students [62]. Complementary to the above, another study reported sex differences in EFs in university students [63].

In this context, another study among university students found that healthy lifestyles (including PA) were positive for the inhibitory control performance [76]. In addition, it has been reported that acute aerobic PA impact positively the inhibitory control [77]. Data from college female students showed that PA was positively related with working memory [78]. This previous evidence contributes to solidifying the positive association between PA and EFs in university students. Complementary to the above, a recent study concluded that increase PA could improve the working memory performance in college students [79]. Building on previously reported findings, it is suggested that during university stage, exercise may be linked to better cognitive flexibility [80]. Similarly, another study indicated that PA was positively associated with EFs, specifically in the executive inhibitory control [9]. Likewise, there is strong evidence about the positive relation between being active and EFs in university students [35]. Consistent with the above, another study has shown that PA are linked to better EFs and increase EFs may improve the academic performance in university students [81].

In this context, an study among university students reported that PA was positively related to cognitive functions such as creativity [86]. Moreover, it has been reported that PA may improve the attention and psychomotor vigilance in undergraduate students [87]. For example, a recent study reported that a single session of aerobic exercise in-crease the BDNF serum levels [88]. Complementary to the above PA interventions have been shown to improve EFs in university students [89]. 

At the same time, however, the sample size is relatively small. 

Response: dear reviewer, the present research is a Pilot study, due to We hope in the future to increase the sample size.

Statistical methods are applied appropriately, with regression coefficients, confidence intervals and significance levels clearly stated. The analysis takes into account multiple variables to provide a comprehensive view of the data.

Response: Thanks for this comment. 

The discussion integrates the findings with existing contemporary literature, highlighting the study's contribution to understanding the effects of physical activity on cognitive function.

Response: Thanks for this comment. 

The conclusions are drawn taking into account some of the limitations of this study. Thus, the study presented in the article is well-conducted and makes a significant contribution to understanding the relationship between lifestyle factors and cognitive function among university students.

Response: Thanks for this comment. 

In the future, it would be appropriate to apply methodological approaches that will enhance the reliability and practical applicability of the results.

Response: We hope in the future to increase the sample size and improve what was requested.

Reviewer 2 Report

Comments and Suggestions for Authors

The manuscript entitled "Healthy lifestyle related to executive function in Chilean university students: A pilot study" delves into a pertinent topic, meticulously examining the interaction between executive function (EF) and lifestyle parameters (i.e., physical activity (PA), sleep duration, screen time (ST), and eating habits) among Chilean university students. However I would like to provide some suggestion in order to improve the article: 

In the introduction section it would be pertinent to articulate the objectives of the study more clearly at the beginning in order to facilitate the understanding of the trajectory and intention of the study from the beginning.

Throughout the text, the expression "Complementary to the above" is used repeatedly, which could seem somewhat monotonous, it is advisable to use synonyms. 

 The methodology is described comprehensively and thoroughly, but I would like to see more data on the sample collection. 

The presentation of the results is quite clear, however in the discussion section I missed comparisons with previous studies in more detail. 

Author Response

Reviewer 2

The manuscript entitled "Healthy lifestyle related to executive function in Chilean university students: A pilot study" delves into a pertinent topic, meticulously examining the interaction between executive function (EF) and lifestyle parameters (i.e., physical activity (PA), sleep duration, screen time (ST), and eating habits) among Chilean university students. However I would like to provide some suggestion in order to improve the article: 

Response: thank you very much fro your suggestion and for the opportunity to improve the paper quality.

In the introduction section it would be pertinent to articulate the objectives of the study more clearly at the beginning in order to facilitate the understanding of the trajectory and intention of the study from the beginning.

Response: dear editor we have adapted the nomenclature and adapting the objective according terms abbreviations  (i.e., PA, ST). The objective now is:

The objective of this study was, therefore, to investigate the association between EFs and lifestyle parameters (i.e. PA, sleep duration, ST and food habits) among Chilean university students and then to determine the differences in the EFs and lifestyle parameters according to sex.

Throughout the text, the expression "Complementary to the above" is used repeatedly, which could seem somewhat monotonous, it is advisable to use synonyms. 

Response: dear reviewer, we have changed this expression.

 The methodology is described comprehensively and thoroughly, but I would like to see more data on the sample collection. 

Response: We hope in the future to increase the sample size and improve what was requested.

The presentation of the results is quite clear, however in the discussion section I missed comparisons with previous studies in more detail. 

Response: thanks you for your comment, we have added information in all introduction.

 In additions, it has been found that there were sex differences in inhibitory control among university students [62]. Complementary to the above, another study reported sex differences in EFs in university students [63].

In this context, another study among university students found that healthy lifestyles (including PA) were positive for the inhibitory control performance [76]. In addition, it has been reported that acute aerobic PA impact positively the inhibitory control [77]. Data from college female students showed that PA was positively related with working memory [78]. This previous evidence contributes to solidifying the positive association between PA and EFs in university students. Complementary to the above, a recent study concluded that increase PA could improve the working memory performance in college students [79]. Building on previously reported findings, it is suggested that during university stage, exercise may be linked to better cognitive flexibility [80]. Similarly, another study indicated that PA was positively associated with EFs, specifically in the executive inhibitory control [9]. Likewise, there is strong evidence about the positive relation between being active and EFs in university students [35]. Consistent with the above, another study has shown that PA are linked to better EFs and increase EFs may improve the academic performance in university students [81].

In this context, an study among university students reported that PA was positively related to cognitive functions such as creativity [86]. Moreover, it has been reported that PA may improve the attention and psychomotor vigilance in undergraduate students [87]. For example, a recent study reported that a single session of aerobic exercise in-crease the BDNF serum levels [88]. Complementary to the above PA interventions have been shown to improve EFs in university students [89]. 

Reviewer 3 Report

Comments and Suggestions for Authors

The theme of the work is relevant. The abstract indicates the objective of the work

The method section is well structured.  Especially design, sampling and result parts are consistent with each other.

I think the manuscript meets expectations, well elaborated and structured. Some modifications are recommended to increase the final quality:

The  material, method and result section needs to be improved;

Information about questionnaires should be given in detail, for example; who developed, who adapted, and detail information about validity and reliability studies

150 university students is so small for cross sectional survey design for generalizability, what about sampling  technique and population.   

 Why researchers conducted ANOVA instead of independent sample t test to determine gender difference. Could you please report your analysis report APA format ( t scores or F scores and  df values as well as p values)

The discussion and conclusions and references section well designed and structured

Comments on the Quality of English Language

Minor editing of English language required

Author Response

Reviewer 3

The theme of the work is relevant. The abstract indicates the objective of the work The method section is well structured.  Especially design, sampling and result parts are consistent with each other. I think the manuscript meets expectations, well elaborated and structured. Some modifications are recommended to increase the final quality:

Response: thank you very much fro your suggestion and for the opportunity to improve the paper quality

The  material, method and result section needs to be improved;

Information about questionnaires should be given in detail, for example; who developed, who adapted, and detail information about validity and reliability studies

Response: we have added to complement the information in yellow in text:

To determine the cognitive domains and EFs, the 40-minute CogniFit neurocognitive assessment battery was used [31, 32]. In this study, apart from the cognitive dimension score (i.e. attention, perception, reasoning, coordination and memory), the EFs (i.e. inhibition, working memory and cognitive flexibility) were examined. Correspondingly, this battery has been reported to have good reliability [32]. In addition, this cognitive battery was previously performed by adult subjects [33].

The application of this evaluation is simple and intuitive, so that any professional can apply it without difficulty. In addition, it is designed so that it can be used both face to face in a consultation or laboratory, and remotely from the homes of patients or participants. This neuropsychological test is carried out online, with an approximate duration of 30-40 minutes. At the end of the evaluation, a complete results report is automatically obtained with the user's neurocognitive profile. In addition, it provides valuable information that, as professionals, can help us detect if there is a risk of any dis-order or problem, recognize its severity, and identify the most appropriate support strategies for each case. It is recommended to perform this neuropsychological assessment when you want to know the brain functioning or cognitive, physical, psychological or social well-being of the patient or participant. This evaluation battery should be used in a complementary way to the professional diagnosis and not as a substitute for the clinical interview. Each of the neuropsychological tasks contained in the Cognitive Assessment Battery (CAB) for professionals has been validated following the scientific method. This ensures appropriate psychometric characteristics for an effective evaluation of the patient's or participant's cognitive status. The cognitive profile of the neuropsychological report has high reliability, consistency and stability. Through cross-sectional research designs, psychometric statistics have been obtained with values close to .9, such as Cronbach's Alpha coefficient. The Test-Retest tests have obtained values close to 1, which demonstrates high reliability and precision [32,33].

According diet we have added:

The original survey classified respondents into “healthy eating” (≥ 13 points), “you are on the right track, you need to improve” (between 10 and 12 points), “unhealthy eating” (between 7 and 9 points) and “unhealthy diet” (≤ 6 points), but, to facilitate data analysis, the first two categories were merged. The final instrument was subjected to validation through expert judgment, carried out by two nutritionists. In addition to the survey, the place of residence and daily meal times on campus were consulted.

 PA we have added

The questions will ask about the time you spent being physically active in the last 7 days. It requests the following from the participants: Please answer each question even if you do not consider yourself an active person. Please think about the activities you do at student time or work, as part of your home or garden tasks, moving from one place to another, or in your free time for recreation, exercise or sport. Think about all the intense activities you did in the last 7 days. Intense physical activities refer to those that involve intense physical effort and make you breathe much harder than normal. Think only about those physical activities that you did for at least 10 minutes straight.

ST and sleep quality we have added:

The ST was computed by the sum of the two questions and analyzed in hours/day. Complementary to the above, a study conducted in a nationwide sample of youth in-dicated that the questions on ST and sleep are applicable in a young population [50]. 

Finally we explain:

The questionnaires and the CogniFit neurocognitive assessment battery were completed individually and in the presence of assistant researchers (who respected data confidentiality and clarified any potential doubts or questions). All the evaluations took place in a computer lab during the morning.

150 university students is so small for cross sectional survey design for generalizability, what about sampling technique and population.   

Response: Dear reviewer, the study is a pilot, where we seek to see the response and see the first relationships according to the lifestyle and cognitive functions, therefore the sample size was according to university students’ volunteers who wanted to participate in this Pilot Study and by convenience. We hope in the short term to increase the sample according your request and estimate the sample size statically. We have added to the text: The sample was intentional and non-probabilistic by convenience.

 Why researchers conducted ANOVA instead of independent sample t test to determine gender difference. Could you please report your analysis report APA format ( t scores or F scores and  df values as well as p values)

Response: dear reviewer, thank you for your comments, we committee a mistake in the text, we used T-test. We have changed by:

Differences in the comparison between the sexes were established using an analysis of variance Student's t-test.

We have added f-value according your request, however according journal format (MPDI) we do not have added df. sorry about that

Table 1. Comparison of study variables according to sex

Total

(n=150)

Female

(n=94)

Male

(n=56)

P-value

F-value

Age (y)

21.61 ± 3.08

21.28±3.15

22.18±2.90

0.083

3.05

Cognitive outcomes

Attention (score)

471.1±124.40

450.45±123.84

506.62±118.31

0.011

6.72

Coordination (score)

362.61±145.54

346.64±153.04

390.08±128.49

0.093

2.86

Reasoning (score)

397.13±162.12

384.91±169.76

418.16±147.34

0.250

1.33

Memory (score)

294.19±158.79

270.34±146.20

335.22±172.26

0.021

5.45

Perception (score)

343.91±98.28

335.94±97.26

357.62±99.49

0.216

1.54

Inhibition (score)

406.90±233.12

410.92±241.62

400.00±219.91

0.793

0.07

Working memory (score)

239.68±191.84

213.56±171.72

284.62±216.75

0.037

4.45

Cognitive flexibility (score)

426.13±235.00

410.01±241.97

453.86±222.14

0.296

1.10

Lifestyle parameters

Diet quality (score)

7.37±3.17

6.96±3.00

8.07±3.36

0.037

4.41

Physical activity (h/week)

2.01±1.76

1.66±1.38

2.61±2.14

0.001

10.83

Screen time (h/day)

2.63±0.78

2.51±0.71

2.84±0.84

0.012

6.44

Sleep (h/day)

6.89±1.52

6.93±1.61

6.81±1.36

0.036

0.23

The data are presented as means and standard deviations, with statistical signifi-cance set at p < 0.05.

The discussion and conclusions and references section well designed and structured

Response: thank you for this comment.

Reviewer 4 Report

Comments and Suggestions for Authors

In detail, I read and analysed the manuscript Healthy Lifestyle Related to Executive Function in Chilean University Students: A Pilot Study.

The topic is current due to the increased attention to healthy lifestyles and their impact on the overall health of young people worldwide. If we consider that the study’s authors examined the influence of healthy lifestyles on the executive functions of students, which are especially developed at this age, then the importance of this study is even greater.

However, the presentation of the research results requires significant corrections.

Abstract

An abstract is concise and well-written.

Introduction

In a very understandable and concise manner, the authors presented facts about executive function and its development in students’ ages. They also analyse healthy lifestyles as public health concerns. At the end of the introduction, they reported the results of previous studies related to the correlation between students’ healthy lifestyle and their executive function.

Relevant sources of information support all the mentioned facts.

Materials and Methods

All used instruments are described in detail. What is unclear, i.e. not stated, should be the method of distributing and collecting questionnaires and how the study’s sample size and response rate were determined. That should be added. The selected and used data analysis methods are appropriate.

Results

Results are presented in five tables, textual, and interpreted appropriately and consistently throughout the manuscript. The table and legend are appropriate and easy to analyse and understand.

Discussion

The authors explained all the results obtained in the discussion. Relevant and actual sources of information support all listed facts.

Potential limitations but strengths of the study are outlined in this section of the manuscript.

Conclusions

The conclusions are stated correctly. 

Author Response

Reviewer 4

In detail, I read and analysed the manuscript Healthy Lifestyle Related to Executive Function in Chilean University Students: A Pilot Study.

The topic is current due to the increased attention to healthy lifestyles and their impact on the overall health of young people worldwide. If we consider that the study’s authors examined the influence of healthy lifestyles on the executive functions of students, which are especially developed at this age, then the importance of this study is even greater.

Response: thank you very much for your suggestion and for the opportunity to improve the paper quality.

However, the presentation of the research results requires significant corrections.

Abstract

An abstract is concise and well-written.

Response: thank you for your comment.

Introduction

In a very understandable and concise manner, the authors presented facts about executive function and its development in students’ ages. They also analyse healthy lifestyles as public health concerns. At the end of the introduction, they reported the results of previous studies related to the correlation between students’ healthy lifestyle and their executive function.

Relevant sources of information support all the mentioned facts.

Response: thank you for your comment.

Materials and Methods

All used instruments are described in detail. What is unclear, i.e. not stated, should be the method of distributing and collecting questionnaires and how the study’s sample size and response rate were determined. That should be added. The selected and used data analysis methods are appropriate.

Response: Dear reviewer, the study is a pilot, where we seek to see the response and see the first relationships according to the lifestyle and cognitive functions, therefore the sample size was according to university students’ volunteers who wanted to participate in this Pilot Study and by convenience. We hope in the short term to increase the sample according your request and estimate the sample size statically. We have added to the text: The sample was intentional and non-probabilistic by convenience.

Results

Results are presented in five tables, textual, and interpreted appropriately and consistently throughout the manuscript. The table and legend are appropriate and easy to analyse and understand.

Response: thank you for your comment.

Discussion

The authors explained all the results obtained in the discussion. Relevant and actual sources of information support all listed facts.

Potential limitations but strengths of the study are outlined in this section of the manuscript.

Response: thank you for your comment.

Conclusions

 The conclusions are stated correctly. 

Response: thank you for your comment
